# A Quest for Livelihood Sustainability? Patterns, Motives and Determinants of Non-Farm Income Diversification among Agricultural Households in Punjab, Pakistan

Muhammad Amjed Iqbal [1], Muhammad Rizwan [2,*], Azhar Abbas [1,*], Muhammad Sohail Amjad Makhdum [3], Rakhshanda Kousar [1,*], Muhammad Nazam [4,*], Abdus Samie [1] and Nasir Nadeem [5]

1. Institute of Agricultural and Resource Economics, University of Agriculture, Faisalabad 38040, Pakistan; amjed.iqbal@uaf.edu.pk (M.A.I.); abdussamie@uaf.edu.pk (A.S.)
2. School of Economics and Management, Yangtze University, Jingzhou 434023, China
3. Department of Economics, Government College University Faisalabad, Faisalabad 38000, Pakistan; sohailmakhdum@hotmail.com
4. Institute of Business Management Sciences, University of Agriculture, Faisalabad 38040, Pakistan
5. Department of Agribusiness and Applied Economics, Muhammad Nawaz Shareef University of Agriculture, Multan 60000, Pakistan; nasir.nadeem@mnsuam.edu.pk
* Correspondence: rizwaneco@yangtzeu.edu.cn (M.R.); azhar.abbas@uaf.edu.pk (A.A.); rakhshanda.kousar@uaf.edu.pk (R.K.); Muhammad.Nazam@uaf.edu.pk (M.N.)

**Abstract:** Many farmers worldwide resort to choosing various income-earning options for diversifying their income sources as a means of risk-avoidance, social protection, and, above all, to finance agricultural operations. Non-farm income generation among farm families has become an imperative part of livelihood earning strategies in recent years amid fast-evolving climatic and sociodemographic changes. In this regard, this study seeks to identify the patterns and socioeconomic factors responsible for the uptake of various non-farm income diversification sources among agricultural households in southern Punjab, Pakistan. For this purpose, a total of 290 farm households were sampled using a random sampling technique to collect relevant data through structured questionnaires. Results show that approximately 79% of the surveyed farmers were involved in non-farm income generation activities, whereas, the income from these sources accounts for about 15% of total household income. The majority of the respondents offered labour for off-farm work followed by self-employment ventures. The major reason to pursue non-farm work includes low income from agriculture, mitigating risks associated with farming, and acquiring funds to finance farming operations, along with the desire to increase family income. A range of socioeconomic and infrastructure-related variables are associated with the decision to participate in specific off-farm activity, such as age, education, family size, farm income, dependency burden, farming experience, and distance to the main city. Results imply the provision of technical support to increase livelihood from farming operations to ensure food security and curb rural-urban migration. However, vocational training can enhance the rural inhabitants' skillset to diversify on the farm through agribusiness development within rural areas, enabling them to employ local people instead of populating urban centres.

**Keywords:** adaptation; sustainability; community support; business; start-up; profitability; constraints

## 1. Introduction

The economy of Pakistan is mainly dependent on agriculture which is the principal source of livelihood for rural people. Since Pakistan has an agricultural economy, economic growth chiefly depends on agriculture as well as farm activities. As a result, agricultural sustainability strongly supports the farmers' livelihood sustainability [1–3]. As Pakistan's agriculture provides employment to almost 40% of the total labour force while having a share of 19% in the country's GDP [4], it is one of the major drivers of livelihood sustainability within the country. Its performance in the last decade has shown a declining

trend due to many factors including climate change, lack of policy focus, poor market integration, farmers' own constraints and preferences in terms of labour availability, financial issues, and alternative working options. Climate change is perceived to be one of the major reasons as farm production is highly dependent on weather conditions beyond human beings' control [5–7]. In addition, recent statistics reveal some good prospects of agricultural growth courtesy of strong institutional support, credit provision, farmers' proactive behaviour, and mass awareness campaigns [8–10].

Declining per capita arable land and water availability as a result of the rapidly increasing population in most parts of the world are the main dilemmas for agrarian economies, including Pakistan [11,12]. Other anthropogenic and institutional drivers also cause an unstable and inconsistent flow of farm incomes, such as droughts, floods, marketing uncertainties, price fluctuations, and incoherent policy environment [13]. Farmers in developed countries also face uncertainty of farm income flows and are tempted to pursue various measures to secure and ensure the uninterrupted flow of income whether through farm-level diversification or off-farm diversification; this includes engaging in part-time businesses, providing labour to other industries or opting to pursue part-time/full-time service employment [14–17]). Such options pave the way for income sustainability among farm families while enabling them to easily and timely finance farm operations. Farmers in Pakistan also face similar issues, along with uncertain policy support, weaker institutional frameworks, higher incidence of crop and livestock disease, rampant pest infestation, higher vulnerability to catastrophic risk, poor resilience and coping capacity, and staggered mode of climate change adaptation [7,18,19]. All these factors place intense pressure on farmers to seek alternative options of income sources other than agriculture.

One must be well-aware and proactive to respond to any risk/risk source to which his/her livelihood is exposed by developing a feasible risk management strategy, for example, pursuing and resorting to non-farm income sources in the case of the farming businesses. Economic motives, as well as the pressure of other risk sources, generally results in persuading the majority of agricultural households to explore other means of generating additional income through non-farm activities to potentially face the issue of income variability and the risk of farm income decline [20] In Pakistan, non-farm activities have become a vital element of income generation strategies among agricultural households with the possibility of increasing their share of non-farm income to total farm income as evinced by [21] and [22], who note that the share of non-farm income in total household income may increase with the passage of time in many developing countries. These activities are also linked with poverty reduction in rural areas [21].

Due to population growth and the uncertain nature of farming, households are pushed towards non-farm activities leading to "distress-push" diversification; this is not the only case as some agricultural households are compelled into non-farm sectors because of higher earnings from off-farm employment. The latter has fewer risk elements than agriculture, resulting in a "demand-pull" diversification [23]. Some studies tacitly accept that the "distress-push" factor is more prevalent due to a declining per capita land availability, thus forcing farm families to pursue non-farm activities [24].

**Dynamics of Non-Farm Income Diversification—Literature Review**

The non-farm sector comprises all non-farm activities that are not directly linked to farm production operations but rather have links to various off-farm enterprises within rural as well as urban and peri-urban areas [22]. These strategies also serve as a method of self-insurance among agricultural households to stabilise their total household income [25]. Generally, the income of agricultural households is a combination of income from farm and non-farm sources. The decline in farm income due to a significant number of farmers (or members of farm families) searching for non-farm employment has increased not only in developing countries but also in developed countries. For example, the Economic Research Service (ERS) data show that income from non-farm sources has become the leading component of farm household income in USA [26].

In the case of the Netherlands, data from 60,000 respondents in [27] show that nearly 40% and 60%of agricultural households received earnings from off-farm services, respectively. Additionally, the sharing of income from off-farm employment gradually rose up to 15% from 2001 to 2013 [28]. Barret et al. [29] show that farmers always try to adopt a range of income sources in which non-farm sources have a leading share. This phenomenon has deeper roots in developing countries, as noted by [30] who discovered that 35–50% of rural farmers' total income comes from non-farm employment, which is anticipated to increase rapidly with the passage of time. These countries share the common issue of an increasing population with stagnant farming sources [20]. In the case of Ghana, [31] indicated that approximately 74% of farmers were engaged in non-farm income generation streams, whereas [32] noted that approximately 75% of agricultural households were involved in off-farm employment opportunities in Taiwan.

Literature reveals that the low risk of investment, more returns, and the usage of extra income from non-farm sources for timely agricultural practices on the farm are the main reasons to adopt off-farm activities [33,34]. It is almost unanimously agreed that farmers with non-farm income are more likely to spend more on seeds, fertiliser, plant protection, and labour [35]. Thus, non-farm income provides a cushion against any uncertain event the farm faces for bearing loss yet allows for the continuity of uninterrupted farming operations [33]. Hence, such variation in vulnerability and risk exposure among various income sources attracts farmers to adopt less-risk associated income options as a risk management strategy [36]. Reardon et al. [37] also noted that households must focus their attention on income diversification strategies to minimize income risks. Consequently, off-farm activities play an important role—apart from reducing risk in income flows—in stimulating the growth of the rural economy and reducing the poverty level [21,38].

Oluwatayo [39], in his study, explored the major determinants of diversification using the tobit regression model for a sample of 420 respondents in the case of rural Nigeria. He discussed the major role of various socioeconomic factors that mainly affect the likelihood of income diversification practices among farmers, including farm families' income from farming, the education level of the household head, and perceived economic situation of the country. Similarly, [40] demonstrated the effect of non-farm income on the living standard of farming households while noting factors including age, patriarchal family structure, formal education level, farm size, and the family's poverty status as major determinants for opting to non-farm income diversification.

The current study is a cornerstone in understanding the patterns and motives of off-farm diversification while contemplating the role of some novel drivers on farmers' decisions to diversify their income. The unique feature of this work comes from its exploration of patterns in off-farm diversification drive, along with the documentation of push and pull factors for such drive. The work takes its theoretical insights from the theory of rational choice (an extension of utility theory), where farmers are assumed to behave rationally and try to increase their utility by expanding the means of their livelihood to find stability and cope with any untoward event [13,41]. The theory assumes that the individual's order of preference depends on the mean and variance of returns from various enterprises or income sources [13]. The farmer then decides among alternate but virtually unlimited possibilities in the enterprise mix to maximize the utility of income coming from various enterprise portfolios. The Modern Portfolio Theory (MPT) equally applies to the concept of off-farm income diversification [42]. Markowitz [43,44] shows how MPT can be applied to reduce risk when many assets are combined together in a single portfolio, but the asset returns are not correlated perfectly. In our study, portfolio diversification stands for a shift away from crop and livestock farming, which is based on (perceived/revealed) risks and returns, to an enterprise or activities mix.

A scant body of literature has explored the pattern, motives, and drivers of non-farm income diversification in the case of Pakistani agriculture. Agriculture has remained the backbone of Pakistan's economy for decades; however, this sector has suffered from multifarious shocks in recent years, casting serious implications for the sustainability

of farm income while forcing select farm communities to leave the farming business altogether [7,19]. These circumstances, however, offer a great avenue for future research, particularly regarding the fate of those who substituted non-farm work for farming as a profession. Nevertheless, the rapid population increase has led to increased pressure on land resources to absorb additional people, and increased demand for food and fibre; this has created a major dilemma for policymakers—in Pakistan and abroad—as to how far switching professions should be promoted/discouraged to create a balance between food provision and sustainable livelihoods. This also offers a challenge for researchers to offer plausible model(s) to incorporate trade-offs linked with switching from farming to other professions. This endeavour, however, is beyond the scope of this article but we entrust it to fellow researchers. With this backdrop, our study was undertaken to evaluate the pattern, motives, and determinants of non-farm employment among farm families of the Punjab province (the largest province in terms of population) who share in the cropped area and overall agricultural production within the country. Moreover, our study also proposes coherent policy options for an effective diversification drive for risk mitigation, climate change adaptation, and rural development.

## 2. Materials and Methods

For the present study, primary data were collected from agricultural households through a multistage random sampling technique. Initially, the Punjab province was selected for the study, which is the most agriculture-dominated province in the country. The province contributes about 53% to the overall agricultural GDP of the country [7] and consists of nine divisions on an administrative basis. Among these nine divisions, three divisions were selected randomly, namely Multan, Bahawalpur, and the DG Khan divisions. From each division, one district was selected randomly, i.e., Bahawalpur, Muzaffargarh, and Vehari. Rural off-farm activities in the study areas were categorised into three types: (i) Services including all types of government jobs and private sector institutions, i.e., teachers, lawyers, village doctors etc., (ii) Self-employment including shop-keeping, pulling haulage carts, driving small transport vans, commissioned agents, fertiliser or pesticide business holders in grain markets, any type of trading or contracting services etc., (iii) Off-farm labour comprising different types of mechanics, daily labour in rural areas, transport operations, construction and similar services either in the vicinity of the village or in the nearby city. A simplified version of farmers' income from various sources is depicted in Figure 1. A proportionate sample from each district was drawn considering the total number of farms in each district. In this regard, 110 farmers were selected from the Bahawalpur district, being relatively larger than the other two. On the other hand, 90 farmers were interviewed from each of the Vehari and Muzaffargarh districts since the number of farms is almost the same within these districts [4]. Thus, the total sample size of this study was 290 farmers.

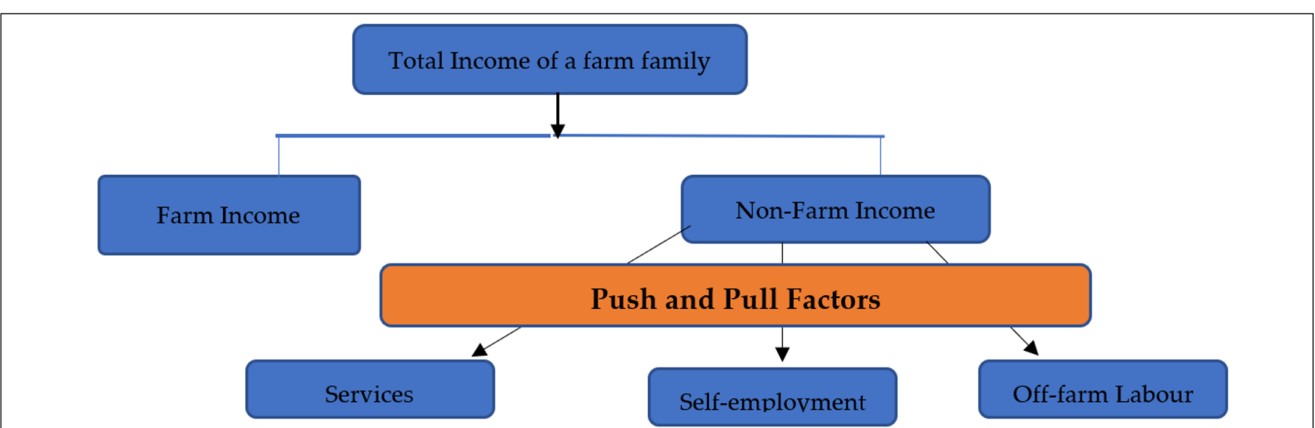

**Figure 1.** Conceptual framework on sources of farmers' income.

Data were collected from agricultural households through in-person interviews due to the high feasibility of this method in the study area. There might have been greater chances of non-cooperation from the respondents if questionnaires were sent through postal or electronic mail. The questionnaire used for data collection contained questions about the socio-economic attributes of respondents, reasons and motives for off-farm employment, and information about factors affecting their choice of off-farm employment as an agricultural household. Before starting the formal survey, a pretesting survey of the designed questionnaire was conducted with 10 farmers who were not included in the final dataset.

We considered participation in off-farm activities as a function of the farm and farmers' characteristics, i.e., age, education, farm size, farm income, distance from main city, etc. The dependent variable for the model was participation in three types of off-farm activities, such as services, self-employment, and off-farm labour. Since the dependent variable has a dichotomous form, the use of the Ordinary Least Square (OLS) model was ruled out. The dichotomous nature of the dependent variable was defined as households participating in off-farm activities receiving a score of 1 and the households engaged in farm activities exclusively receiving a score of 0. Therefore, the binary logistic model was used to estimate the covariates of households' participation in non-farm activities [21,45–47].

The Binary logistic model for this study is indicated as follows:

$$\frac{P_i}{(1 - P_i)} = \frac{1 + \exp(Z_i)}{1 + \exp(-Z_i)} \tag{1}$$

As the above equation is non-linear, it can be linearized by taking the natural log. Hence the model provides assumes the form denoted below:

$$Li = Ln\left[\frac{Z_i}{(1 - P_i)}\right] Z_i = \beta_o + \beta_1 X_{i1} + \cdots + \beta_9 X_{i9} + e \tag{2}$$

where $Pi/(1 - Pi)$ is the ratio of probability that the farmer will be involved in non-farm activities to the probability that the farmer will not be involved in specific non-farm activities. So, the endogenous variable is binary, and has two values of 1 and 0.

$\beta_o$ = constant
$\beta_1 - \beta_9$ = coefficients of logistic regression
$X_{i1}$ = Age of ith respondent where I = 1, 2, 3 . . . 290.
$X_{i2}$ = education (number of years of formal education)
$X_{i3}$ = farm size (total farming area of the respondents in acres)
$X_{i4}$ = location (distance from main city) in kilometres.
$X_{i5}$ = No. of household workers (number of earning members in the household)
$X_{i6}$ = dummy variable for having access to road (1 for yes and 0, otherwise)
$X_{i7}$ = dependency ratio (total number of household members divided by the total number of earning members in the family)
$X_{i8}$ = farming experience of household in years
$X_{i9}$ = family size (total number of family members)
$e$ = error term

## 3. Results and Discussions

### 3.1. Descriptive Analysis of the Studied Variables

The summary of statistics for variables used in the model are presented in Table 1.

**Table 1.** Summary statistics of the variables used for regression.

| Variables | Mean | Standard Deviation | Minimum | Maximum |
|---|---|---|---|---|
| Age | 45.42 | 8.18 | 24.00 | 68.00 |
| Education (Years) | 9.61 | 3.39 | 0.00 | 16.00 |
| Total farming area (acre) | 15.23 | 7.22 | 3.00 | 37.00 |
| Location (Distance from Main city in kilometres) | 11.33 | 4.35 | 3.00 | 30.00 |
| Family size | 8.25 | 1.47 | 4.00 | 12.00 |
| Number of earning Members in family | 2.74 | 0.73 | 2.00 | 4.00 |
| Dependency burden | 3.01 | 1.02 | 1.67 | 6.00 |
| Access to road | 0.94 | 0.24 | 0.00 | 1.00 |
| Farming experience (years) | 20.99 | 8.82 | 2.00 | 45.00 |

Table 1 indicates that the average age of the farmers is approximately 45 years, while the average years of education for the respondents is 9.6 years; this implies that the farmers were generally middle-aged and received their secondary school certificate (Matric) on an average performance basis. The mean distance of the village from the main city is approximately 12 km. Similarly, every family has, in total, eight members, out of which approximately three are the earning members, thus having a dependency burden of approximately three; this implies that each earning member of the family has to feed and support three members of their family including themselves. The majority of agricultural households from the sampled areas had access to roads from their village, and the mean farming experience was approximately 21 years.

*3.2. Patterns, Reasons and Motives of Non-Farm Activities*

The results in Table 2 present the pattern of farmers' activities in the study areas. It is clear that the majority of the farmers were involved in off-farm/non-farm work in one way or another through their involvement in the services sector, self-employment or off-farm labour work. Out of the contacted respondents, 62 (21.4%) farmers resorted only to farming while the remaining 228 farmers took up other professions along with farming activities. The uptake of off-farm labour is most pronounced in the study area, followed by self-employment and services. Rashidin et al. [48] noted that a major part of non-farm income originates from off-farm labour contribution. Using relatively different data and study areas, [49] show that farming is a major source of income among rural farm households. The second major non-farm income source is self-employment, where farmers find it convenient to start local businesses such as inter alia, retail shops, fruit and vegetable stalls, tea stalls, ice selling, snack stalls (baking of *samosas and pakoras*), meat shops, selling animals etc. Table 2 also portrays the majority of farming families in the Bahawalpur district as having their family members involved in the services sector, such as teaching, health, banking industry jobs, and other related services. These decisions are justified based on these jobs' having divisional headquarters and a metropolitan area with relatively larger markets and population compared with the other two districts. Similarly, self-employment is more pronounced in this district mainly because of the reasons recorded earlier. People can scope out customers for any type of vending operation and find the possibility of success in the transport business to and from the city areas. Providing off-farm labour is more common in the two smaller districts viz. Vehari and Muzaffargarh.

**Table 2.** Pattern of non-farm income diversification (percent respondents in each category of non-farm income source).

| Districts | Farming Only | Non-Farm Employment Sources | | | Total (Farming + Non-Farming) |
|---|---|---|---|---|---|
| | | Services | Self-Employment | Off-farm Labour | |
| Bahawalpur | 16.4 | 41.3 | 46.7 | 32.6 | 100 |
| Muzaffargarh | 25.5 | 26.8 | 28.4 | 53.7 | 100 |
| Vehari | 23.3 | 31.3 | 37.3 | 47.7 | 100 |
| Overall | 21.4 | 26.5 | 30 | 33.8 | |

Table 3 identifies possible reasons for the involvement of agricultural households in non-farm employment activities. These reasons were ranked based on the importance assigned to them by the respondents and were analysed using the Z score method [50]. According to the Z score results, respondents ranked low income from agriculture as the topmost reason to participate in non-farm activities. This finding is in agreement with [51,52]. Increasing family income was ranked as the second reason for off-farm employment participation. Farmers affirmed that the burden of a large family was the third topmost reason to become involved in off-farm income activities, while the fourth-ranking was "the availability of off-farm opportunity".

**Table 3.** Ranking of reasons for participation in non-farm employment activities.

| Reasons | Z Score | Rank |
| --- | --- | --- |
| Low income from agriculture | 3.21 | 1 |
| To increase family income | 1.86 | 2 |
| Burden of large family | 1.42 | 3 |
| Availability of off-farm opportunities | 0.75 | 4 |
| Reduction in income risk from farming | 0.33 | 5 |
| Desire to work on something else | 0.29 | 6 |

In a similar vein, the perceived motives for the uptake of non-farm activities are depicted in Figure 2. The majority of farmers opt for non-farm income opportunities to mitigate risks of decreasing farm income. The second frequently reported option was acquiring funds to finance farming operations (about 60% of farmers reported this motive). The least-reported motives included "just to use extra time", "to utilize surplus family labour", and "to maintain status quo".

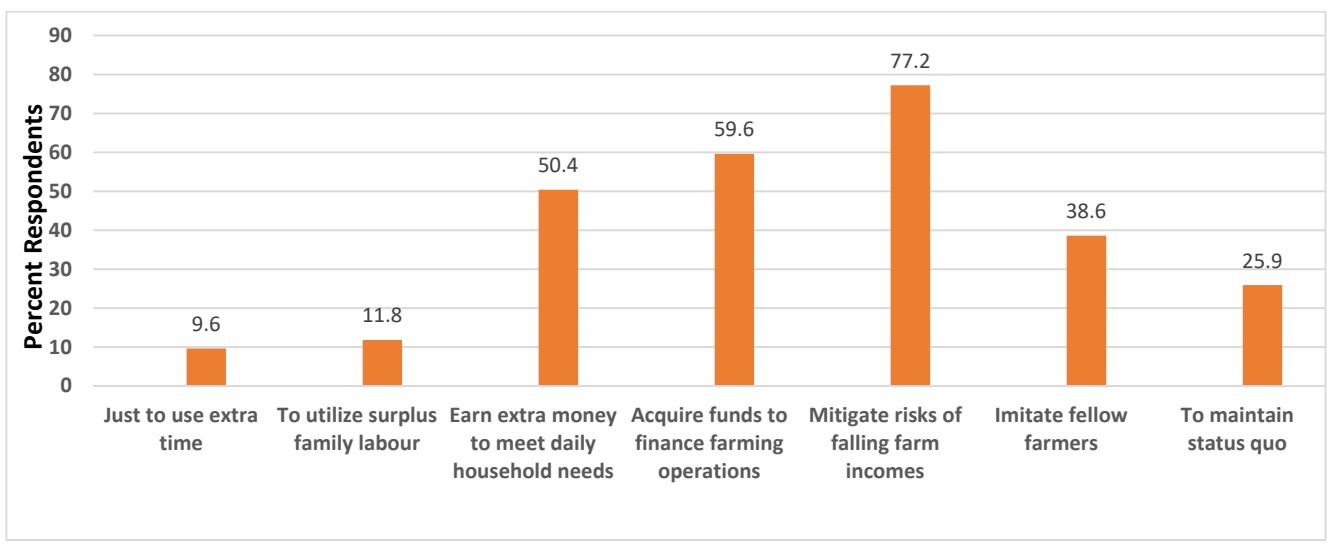

**Figure 2.** Motives behind the uptake of non-farm income opportunities.

### 3.3. Segregation of Farm and Off-Farm Income of the Respondents

Data related to annual income obtained from each crop and farm enterprise were collected from the respondents. Similarly, farmers' household income from off-farm work was also elicited and compiled to create a comparison. Table 4 presents this composition and compares the incomes from farming and non-farm activities among three categories of respondents. It is clear from these findings that farm income is the major contributor, having an approximately 85% share of the total income for the selected farmers from the study areas. Another fact is observed regarding crop income, where the annual income from crops is highest among self-employed farmers, followed by off-farm labour providers

and those employed in services. The opposite is true in the case of livestock income which is highest among off-farm labour providers and lowest among self-employed farmers. On the other hand, there is not a great deal of difference in terms of off-farm income from three income sources, although there is little income difference for those involved in self-employed activities compared with other categories of off-farm work. Self-employed farmers have relatively higher incomes from such activities.

**Table 4.** Average annual income (PKRs) of the respondents from various sources.

| Sources | Self-Employment | Off-Farm Labour | Services |
|---|---|---|---|
| | Farm income | | |
| Crop Income | 566,517.02 | 516,861.95 | 458,969.90 |
| Livestock Income | 56,275.00 | 63,760.00 | 59,624.16 |
| Others * | 3920 | 14,000.00 | 30,100.00 |
| Sub-Total (A) | 626,712.02 | 594,621.95 | 548,694.07 |
| | Off-farm Income | | |
| Self-employment/Business | 118,933.33 | | |
| Off-farm Labour | | 101,040.00 | |
| Services | | | 102,760.00 |
| Sub-Total (B) | 118,933.33 | 101,040.00 | 102,160.00 |
| Total (A + B) | 745,645.35 | 695,661.95 | 650,851.07 |
| Proportion of Off-farm Income to Total Income | 15.9% | 14.2 | 15.7 |

* It includes the income from pension, remittances and other unearned income.

### 3.4. Determinants of Participation in Off-Farm Activities

The parameters of the logit model estimated to identify the elements prompting participation in off-farm activities are presented in Table 5.

Three regression models were used separately for the business, off-farm labour, and service sectors to explore significant factors affecting the participation of farmers in these three types of off-farm activities based on their binary response on participation in such activities.

Table 5 shows the results of binary logistic regression and explains the determinants for agricultural households' participation in any of the self-employed activities. Non-significant Hosmer and Lemeshow (H-L) test results show that the model is good fit. Cox and Snell $R^2$ and Negelkerke $R^2$ values are 0.194 and 0.270, which denotes that the model explains approximately 19–27% of the variations. The model's predictive ability was 72.8%.

The coefficients of the variables do not provide direct information about the effect of changes in explanatory variables on the probability of participation in self-employed activities. Knowing this odds ratio/$Exp(\beta)$ is necessary to discern; this is the ratio of probability for participating in self-employed activities to the probability that the person will not take part. The variables that significantly affect the probability of participation in self-employed activities include the total farming area, number of household workers, access to roads, dependency ratio, farming experience, and family size. Among these total farming areas, the number of household workers, access to roads, and dependency ratio are the variables which positively affect the probability of participation in self-employment options, whereas farming experience and family size have negative signs. The results explain that by increasing the total farming area by one unit, the chances of a farmer pursuing self-employment increases by the value of the associated odds ratio, which is 1.05. This finding is intuitive and justified because with increased farm area, households may receive additional income that can be invested in self-employment options such as opening a local retail shop, setting up vegetable/fruit stalls or starting a taxi company [53].

**Table 5.** Results of logit regression model.

| Variables | Services | | Self-Employment | | Off-Farm Labour | |
|---|---|---|---|---|---|---|
| | β | Odd Ratio | β | Odd Ratio | β | Odd Ratio |
| Age | 0.041 (0.038) | 1.041 | 0.008 (0.038) | 1.008 | −0.025 (0.037) | 0.976 |
| Education | 0.26 ** (0.069) | 1.297 | −0.063 (0.059) | 0.939 | −0.168 *** (0.058) | 0.845 |
| Total farming area | −0.054 ** (0.026) | 0.948 | 0.05 ** (0.025) | 1.051 | 0.008 (0.024) | 1.008 |
| Location | 0.015 (0.041) | 1.015 | 0.06 (0.041) | 1.062 | −0.081 * (0.045) | 0.922 |
| Household Workers | −1.363 (1.036) | 0.256 | 2.814 ** (1.218) | 16.681 | −0.821 (1.001) | 0.44 |
| Access to road (dummy) | 2.278 ** (1.146) | 9.757 | 2.431 *** (0.876) | 0.088 | 1.542 (1.106) | 4.673 |
| Dependency ratio | −1.803 ** (0.918) | 0.165 | 2.872 *** (1.078) | 17.679 | −0.63 (0.863) | 0.532 |
| Farming experience | −0.005 (0.033) | 0.995 | −0.084 ** (0.037) | 0.919 | 0.061 ** (0.032) | 1.063 |
| Family Size | 0.727 * (0.401) | 2.069 | −1.294 ** (0.511) | 0.274 | 0.293 (0.386) | 1.34 |
| Farm income | 0.031 (2.38) | 0.92 | 0.066 ** (2.57) | 1.68 | 0.028 (0.66) | 1.80 |
| Constant | −3.116 (3.355) | 0.044 | −4.271 (3.353) | 0.014 | 1.819 (3.054) | 6.163 |
| Model Prediction Success (MPS) | 76.1% | | 72.8% | | 72.8% | |
| Log-likelihood ratio | 192.327 | | 190.299 | | 200.061 | |
| Hosmer and Lemeshow Test | (df = 8) significance test result 11.474 (*p*-value = 0.176) | | (df = 8) significance test result 7.635 (*p*-value = 0.470) | | (df = 8) significance test result 2.833 (*p*-value = 0.944) | |
| Cox and Snell R$^2$ | 0.185 | | 0.194 | | 0.149 | |
| Negelkerke R$^2$ | 0.257 | | 0.270 | | 0.207 | |

* Significance at 10%; ** Significance at 5%; *** Significance at 1%.

Similarly, the probability of participating in self-employed activities increases with both the rising number of household workers and the dependency ratio. It is understandable that as household workers and dependency ratio increases, the higher the probability that other members will pursue these types of activities. This finding reflects the overall social fabric and households' intent to counter additional pressure on farm sources to feed family members. They can find ways to manage this burden by sending a few family members (increased family size causes the dependency burden to rise as well) to work off-the farm for additional income. On the other hand, the rising dependency ratio means more family members become dependent on working members. Here the choice is none except to find extra time to work off- farm to support dependent household members. In this scenario, there will of course be trade-offs between farm output and additional income from self-employment activities. On the contrary, farming experience and family size decrease the probability of self-employed activities. This finding on farming experience is supported by the fact that increased experience provides an added advantage to the farmer for increased farm output by employing his/her skill to obtain higher agricultural

earnings [7]. The sign of the variable on family size is unexpectedly negative, as reported by [54]. There may be time constraints as larger households may have more children or elderly in their family, therefore more time is required for their care. Makate et al. [55] have also shown similar a relationship to farming experience with diversification in Zimbabwe. However, the outcome of the increase in family size variable is against our a priori expectation. In a developing country context, [56] showed a negative impact on family size for off-farm diversification in Southern Ethiopia. The reason for such a relationship can be the added pressure on the household head to create on-farm employment for additional family members instead of sending them to work off-farm [13]. The farmers with better access to roads have more chances of accomplishing self-employment. These results are expected because having more infrastructure accelerates the probability of pursuing self-employment.

Another regression was performed using the dependent variable on off-farm labour with the same independent variables. Results show that the Hosmer and Lemeshow test (H–L) values are nonsignificant, indicating the model is a good fit, and the values of Cox & Snell $R^2$ and Negelkerke $R^2$ indicate that 14–20% variations are explained by the model with an MPS value of 72.8%. The probability of participating in off-farm labour activities decreases with education and location. Dary and Kunnibe [47] and [53] also reported similar findings. Educated farmers located far from the main city stand at a 0.845 and 0.922 chance of not pursuing off-farm labour activities. In fact, people with more education prefer to not pursue labour activities in the current social setup. It is also difficult for people residing in remote areas to travel to city areas and find labour opportunities. However, more experienced farmers are more likely to participate in labour activities. Usually, small landholding farmers try to find labour activities to obtain more income in addition to farm income.

The last regression results exposed factors inflating farmers' behaviour to adopt off-farm activities. The results of this model depict that it is a good fit as expressed by the (H–L) test values, and 18 to 25% variations are explained by the model, which is represented by Cox & Snell $R^2$. The model has the highest MPS value, which is 76.1%, as compared with the other two regression models provided. In the case of service jobs, education, access to roads, and family size played a positive and significant role, while total farming area and dependency showed a negative impact. The odds increase by 1.297 for educated people pursuing services. According to [57], more educated households of rural populations have more right of entry to employment opportunities. De Janvry and Sadoulet [58] also illustrated that education is one of the main elements for pursuing the non-farm sector. Similarly, farmers with a large family size and more access to roads are more likely to take part in off-farm service jobs [59]. The odds for education, access to roads, and family size increases about 1.297, 9.757, and 2.069, respectively. However, the odds for total farming area and dependency ratio decreases by about 0.948 and 0.165.

## 4. Conclusions, Policy Recommendations, and Future Outlook

With the passage of time, it has become crucial for the majority of the farmers to espouse non-farm economic activities to fulfil farming expenses and increase their livelihood. Due to climate change and other reasons, exclusive dependence on agricultural income among the farming community has become laden with heavy risk due to its uncertainty, hence farmers try to support their income from non-farm sources. We divided non-farm income sources into three types for our research, i.e., services, off-farm labour, and self-employment. In order to gain analytical insights, primary data from 290 farm households were collected through random sampling technique using structured questionnaires from three districts: Muzaffargarh, Vehari, and Bahawalpur. Logistic regression analysis was used to discover factors affecting the respondent's choice of non-farm income sources. The study also evaluated the pattern of off-farm income sources among studied respondents along with various motives and reasons to opt for such activities. Various household and geographic factors were assessed for their possible impact on the choice of diversification.

The logit results showed that total farming area, number of household workers, and dependency ratio had a significant and positive impact on endorsing non-farm activities like self-employment. The probability of participation in off-farm labour provision increased with farming experience. However, the likelihood of participation in service jobs or off-farm activities increased with higher education levels, better access to roads, and family size. The study findings imply a major focus on streamlining diversification options among farming communities by providing them relevant information about various job openings, nature of work, and possible payoffs. Moreover, since education has a significant impact on the uptake of various off-farm activities, providing of vocational and technical education and enhancing skillsets of the rural masses would be beneficial in a symbiotic manner by complementing their farming operations through vis-à-vis off-farm activity. These complementarities arise from the fact that farmers with better skills can execute timely and effective decisions about crop choice, input substitution, and output market targeting, thereby leading to increased farm earnings in the same farm area; this would, in turn, lead to livelihood sustainability, more disposable personal income and greater access to diversified and nutritious food (apart from farm-grown traditional food). In this scenario, they are destined to enjoy a sense of food security, additional household amenities, better personal and child health outcomes, and above all, gender mainstreaming. The gendering of livelihood comes from the fact that women find additional choices for leisure, education, employment, and secured social status once livelihood sustainability is achieved (or is perceived to be achieved in the near future). Therefore, they can also leave home for off-farm work instead of exclusively attending to domestic duties and then striving to participate on the farm and accompany the men. These activities have been shown (rather are evident) to affect women's health, ultimately impacting their childbearing and child-raising capabilities.

Off-farm income diversification through services and self-employment is also shown to be significantly related to road access, implying the significance of infrastructure. Improved infrastructure would be helpful in many ways for the rural inhabitants in their drive for on- and off-farm diversification. Better road access is a prerequisite for easy commutation between localities for professional and service-related matters, thereby saving sufficient time that the farming community can devote to other purposes to earn supplementary livelihoods.

Limitations and Future Outlook

The current work was limited to three districts and a data set of 290 farmers. Therefore, the results cannot be potentially generalised for a larger study area. Future work can be geared towards incorporating a larger sample size from a diverse study area for better policy insights.

In order to conduct future research on the topic, one must understand the trade-offs between on-farm diversification vs. off-farm diversification along with the trade-offs in terms of potentially reduced farm output due to the involvement of family members in off-farm work. In a similar vein, trade-offs also need to be quantified in switching from farming to other professions. One also needs to perform a segregated analysis of off-farm diversification among small and commercial tillers, whereas an account on the evaluation of policy interventions, institutional support, and social norms supporting off-farm diversification must be presented. The role of women in the household in the uptake of off-farm diversification is largely missing and has the potential to yield interesting insights and policy relevance.

**Author Contributions:** Conceptualization, M.A.I. and A.A.; methodology, M.R.; software, R.K.; validation, M.S.A.M., M.N. and A.A.; formal analysis, M.A.I., A.A., N.N.; investigation, R.K., A.S.; resources, A.S., R.K.; data curation, A.A., A.S.; writing—original draft preparation, M.R.; writing—review and editing, R.K., N.N.; visualisation, M.R.; supervision, M.A.I.; project administration, A.A.; funding acquisition, M.R., A.A. All authors have read and agreed to the published version of the manuscript.

**Funding:** This research received funding form Yangtze University Doctoral Initiative Project Fund for APC payment.

**Institutional Review Board Statement:** Ethical review and approval were waived for this study, due to the nature of the information being non-sensitive.

**Informed Consent Statement:** Informed consent was obtained from all subjects involved in the study.

**Data Availability Statement:** The data for reported results can be made available upon request for a research purpose.

**Acknowledgments:** The authors want to acknowledge the support provided by field staff in collecting data as well as the study respondents who provided the required information cordially. Authors would also like to thank three anonymous reviewers for their constructive remarks on the earlier versions of the manuscript.

**Conflicts of Interest:** The authors declare no conflict of interest.

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
