# Peer review of "A Quest for Livelihood Sustainability? Patterns, Motives and Determinants of Non-Farm Income Diversification among Agricultural Households in Punjab, Pakistan"

_sustainability, doi:10.3390/su13169084_

Round 1

Reviewer 1 Report

This is an interesting and well-written paper. All parts are extensively described and the methodology is clear.

I have only few minor remarks:

- the authors contribution could be stated more clearly. Although you have provided an extensive literature review part, you could highlight your contribution in straightforward way: please write what make your research unique

- At optimum you could refer to some economic theories that could serve as a background of your research  

- I understood that you run your econometric model for all 290 farms at the same time. The possible extension of econometric modelling would be to add additional covariate which is “belonging to a certain district”. It could be that if farm is located in one district then it is more likely that this farm would be interested in specific non-farm activity. It may be caused by the specific socio-economic situation in a given district that we don’t know  

- the policy recommendation part could be expanded. You could also add some limitations of your research or possible fruitful line for further research

- line 29 – the word „go” or similar is missing

- l. 53 – please, avoid phrases such as „so on”

Reviewer 2 Report

The paper is interesting and properly organized. The research problem has been successfully solved. The tested features were selected properly. The paper has a high cognitive value in terms of empirical research. However, I have a few remarks that require clarification or more extensive writing in the text. Firstly, before presenting the research results, it would be valuable to describe the problem of insufficient income of agricultural farm members in various parts of the world, as well as the ways of solving this problem in the context of driving towards sustainable agriculture, and then to focus on the specificity of agriculture and the national economy in Pakistan. This will help the readers understand the phenomenon under investigation better. Secondly, the limitations of the research resulting from the relatively small size of the data collection need to be clearly stated. Thirdly, if the authors are deciding to formulate recommendations for a policy, it will be worth pointing to the suggested specific directions and actions.

Reviewer 3 Report

I would like to thank the authors for this interesting work.

The research seeks to identify the pattern and socioeconomic factors in the uptake of various non-farm income diversification sources among agricultural households in Southern Punjab, Pakistan.

The research objective as it is presented is quite limited. Exploring patterns and socioeconomic factors is interesting, however, the most important is to make valuable recommendations.

The researchers need to add another section for discussing the results and making sound recommendation for policy makers.

The abstract contains interesting results: "Results imply the provision of technical support to increase livelihood from farming operations in order to ensure food security and to curb rural urban migration. However, vocational trainings can enhance skill-set of the rural inhabitants to diversify on the farm through agribusiness development within the rural areas enabling them to provide employment to local people instead of populating urban centers".

Ideas like technical support, livelihood, food security, vocational training are highly interesting and need to be discussed over the discussion and conclusion section.

The literature review section and research methodology are adequately presented.

Round 2

Reviewer 3 Report

I would like to thank the authors for incorporating the necessary changes. The quality of the manuscript is optimal and deserves to be published.